# Balanced Viscoelastic Properties of Pressure Sensitive Adhesives Made with Thermoplastic Polyurethanes Blends

**DOI:** 10.3390/polym11101608

**Published:** 2019-10-03

**Authors:** Mónica Fuensanta, María Agostina Vallino-Moyano, José Miguel Martín-Martínez

**Affiliations:** 1Adhesion and Adhesives Laboratory, University of Alicante, 03080 Alicante, Spain; monica.fuensanta@ua.es; 2National University of Tucumán, San Miguel de Tucumán, Tucumán T4000, Argentina; agovallino@gmail.com

**Keywords:** thermoplastic polyurethanes blends, pressure sensitive adhesives, viscoelastic properties, adhesion properties, tack, creep, cohesion properties

## Abstract

Pressure sensitive adhesives made with blends of thermoplastic polyurethanes (TPUs PSAs) with satisfactory tack, cohesion, and adhesion have been developed. A simple procedure consisting of the physical blending of methyl ethyl ketone (MEK) solutions of two thermoplastic polyurethanes (TPUs) with very different properties—TPU1 and TPU2—was used, and two different blending procedures have been employed. The TPUs were characterized by infra-red spectroscopy in attenuated total reflectance mode (ATR-IR spectroscopy), differential scanning calorimetry, thermal gravimetric analysis, and plate-plate rheology (temperature and frequency sweeps). The TPUs PSAs were characterized by tack measurement, creep test, and the 180° peel test at 25 °C. The procedure for preparing the blends of the TPUs determined differently their viscoelastic properties, and the properties of the TPUs PSAs as well, the blending of separate MEK solutions of the two TPUs imparted higher tack and 180° peel strength than the blending of the two TPUs in MEK. TPU1 + TPU2 blends showed somewhat similar contributions of the free and hydrogen-bonded urethane groups and they had an almost similar degree of phase separation, irrespective of the composition of the blend. Two main thermal decompositions at 308–317 °C due to the urethane hard domains and another at 363–373 °C due to the soft domains could be distinguished in the TPU1 + TPU2 blends, the weight loss of the hard domains increased and the one of the soft domains decreased by increasing the amount of TPU2 in the blends. The storage moduli of the TPU1 + TPU2 blends were similar for temperatures lower than 20 °C and the moduli at the cross over of the moduli were lower than in the parent TPUs. The improved properties of the TPU1 + TPU2 blends derived from the creation of a higher number of hydrogen bonds upon removal of the MEK solvent, which lead to a lower degree of phase separation between the soft and the hard domains than in the parent TPUs. As a consequence, the properties of the TPU1 + TPU2 PSAs were improved because good tack, high 180° peel strength, and sufficient cohesion were obtained, particularly in 70 wt% TPU1 + 30 wt% TPU2 PSA.

## 1. Introduction

Pressure sensitive adhesives (PSAs) are polymeric materials that can form an immediate bond without chemical reaction to a substrate upon brief contact by applying light pressure for short time [1]. Typical applications of PSAs include labels, sticky notes, packaging, diapers, auto/masking tapes, bandages, and decals. PSAs form physical bonds at a molecular level and sustain a minimum level of stress upon de-bonding. PSAs are soft and viscoelastic solids which have properties derived from the differences in the energy gained in forming van der Waals interactions with a substrate and the energy dissipated during the de-bonding.

Most PSAs are based on acrylics [2] and natural and synthetic rubbers [3], and, less commonly, silicones [4] and polyurethanes [5]. Rubber PSAs are composed of rubber polymer and low molecular-weight compatible tackifier, whereas acrylic PSAs are composed of mixtures of random acrylic copolymers of long side- and short-side chains, acrylic acid, and a tackifier can also be added. Silicone PSAs are used mainly when low temperature use or high-temperature stability is required and they do not contain tackifier [4]. Table 1 summarizes some properties of the typical polymers used as PSAs. Rubber PSAs show excellent tack, peel strength, and cohesive strength, but they have poor skin sensibility and low skin trauma, and low solvent resistance. On the contrary, acrylic and silicone PSAs show low to medium tack and peel strength but good solvent resistance. On the other hand, polyurethane PSAs show adequate resistance to the temperature and the solvents, and they have better low temperature performance than the acrylic or rubber PSAs. Nevertheless, polyurethane PSAs are limited by their low tack and low peel adhesive strength (Table 1).

Medical tapes are one of the most exigent products based on PSAs because of the need to be compatible with the skin [6,7,8]. Acrylic PSAs used for medical tapes produce irritation of the skin due to the presence of unreacted monomers and residual solvent. Despite natural and synthetic rubber PSAs usually being employed for skin contact, their formulations contain tackifiers and additives that produce irritability and skin trauma with long-time application [9]. Silicone PSAs are preferred in medical tapes due to lower skin trauma, good biocompatibility, and low toxicity, but they have high cost [8,10]. Similar to silicone PSAs, polyurethane PSAs show high compatibility to the skin but they are less costly; however, they showed low tack and peel strength.

The performance of PSAs is tightly related to their viscoelastic properties, i.e., the viscous component (to wet the substrate for good contact during bonding) and the elastic component (to withstand shear stresses and peel forces during de-bonding). Copolymers with segmented structure are commonly used for controlling the viscoelastic properties of the PSAs because the hard phase imparts the elastic properties and the soft phase imparts the viscous properties [11]. In this sense, thermoplastic polyurethanes (TPUs) are potential versatile polymers for manufacturing PSAs because of their segmented structure, comprised of hard and soft segments which are thermodynamically incompatible, leading to microphase separation; on the other hand, the hydrogen bond interactions between the hard segments determine the final morphology of the TPUs [12]. Unfortunately, polyurethane PSAs are limited because of their inherent low tack and low peel adhesive strength [13], i.e., the pressure-sensitive adhesion property is not typical of polyurethanes. For improving the tack of the polyurethane PSAs, tackifier resins and/or plasticisers can be added to increase the glass transition temperature (*T*_g_) and decrease the modulus at room temperature [14,15,16]. Different tackifier resins (rosin esters, coumarone-indene resins, unsaturated aliphatic hydrocarbon resins, polyterpene resins) have been added in polyurethane PSAs, and even they displayed high peel strength, the substrate can be damaged during de-bonding due to the migration of the tackifier resin to the surface with time. Additional strategies have been proposed for balancing the properties of the polyurethane PSAs. Thus, the use of low NCO/OH ratios, i.e., insufficient equivalents of isocyanate with respect to the equivalents of high functionality polyol, has been proposed but the low degree of cross-linking in the polyurethane PSAs produced poor cohesion [17,18,19,20]. Nakamura et al. [21] have shown that the addition of one crosslinking agent increased the peel strength of polyurethane PSAs but the tack did not increase sufficiently. On the other hand, more recently, thermoplastic polyurethanes (TPUs) with pressure sensitive adhesion properties (TPU PSAs) with good tack but insufficient peel strength and poor cohesion have been synthesized by reacting 4,4-diphenylmethane diisocyanate (MDI) with 1,4-butanediol chain extender and mixtures of polypropylene glycols (PPGs) of different molecular weights (1000 and 2000 Da) [22]. TPU PSAs synthesized with mixtures of PPGs containing 50 wt% or more PPG of higher molecular weight showed good tack at 10–37 °C and their pressure sensitive adhesion was related to their minor content of bonded urethane groups and important degree of phase separation [22]. Furthermore, these TPU PSAs followed the Dahlquist criterion and they showed low glass transition temperatures, but they had low cohesion and low 180° peel strength.

For balancing the adhesion and cohesion properties, in a recent approach, TPU PSAs with different hard segments content were synthesized by using different mixtures of PPGs with molecular weights of 450 and 2000 Da [23]. The hard segments contents and the degrees of phase separation of the TPUs affected their pressure sensitive adhesion properties. The increase in the hard segments content increased the percentage of the hydrogen bonded urethane groups and produced a lower degree of phase separation in the TPUs, the storage moduli of the TPUs increased, and high shear PSAs were obtained. On the other hand, TPU PSAs with lower hard segments content showed high tack and adequate de-bonding properties, whereas the increase in the hard segments content increased the cohesive strength, the storage moduli of the TPUs, and the 180° peel strength values. Despite the change in the hard segments content of the TPUs producing adjustable properties in the PSAs, an adequate balance of tack, cohesion, and adhesion was not achieved, i.e., the TPU PSAs showing high tack and sufficient peel strength had low cohesion and the ones with low tack and peel strength showed high cohesion.

In this study, TPU PSAs with satisfactory tack, cohesion, and adhesion were prepared. A simple procedure for balancing the adhesion and cohesion of the TPU PSAs was used, consisting of the physical blending of two TPUs with very different properties, i.e., one TPU with excellent tack and poor cohesion (TPU1) and another TPU with good cohesion and poor tack (TPU2). For preparing the TPUs PSAs, the TPUs were dissolved in methyl ethyl ketone (MEK) and the solutions were mixed; blends of TPU1 and TPU2 containing 60–80 wt% TPU1 were prepared and characterized.

## 2. Materials and Methods

### 2.1. Materials

4,4′-Diphenylmethane diisocyanate (MDI) flakes—Desmodur^®^ 44MC (Covestro, Leverkusen, Germany)—was used. Polypropylene glycols with molecular weights of 450 Da (PPG450), Alcupol^®^ D0511, and 2000 Da (PPG2000), Alcupol^®^ D2021, both supplied by Repsol (Madrid, Spain), were used as polyols. Before use, the polyols were melted and dried at 80 °C under reduced pressure (300 mbar) for 2 h. 1,4-butanediol (BD) was used as a chain extender and dibutyl tin dilaurate (DBTDL) was used as a catalyst, both were supplied by Sigma Aldrich Co. LLC (St. Louis, MO, USA). Methyl ethyl ketone (MEK) (Jaber Industrias Químicas, Madrid, Spain) was used as the solvent for the TPUs.

### 2.2. Synthesis of the Thermoplastic Polyurethanes (TPUs)

The thermoplastic polyurethanes (TPU1 and TPU2) were synthesized using the prepolymer method and an NCO/OH ratio of 1.10 was selected. The polyurethane prepolymer was synthesized in a 500 cm^3^ four-neck round-bottom glass reactor under nitrogen atmosphere (flow: 50 mL/min) by reacting the melted MDI with the mixtures of PPGs under stirring with an anchor shaped stirrer in a Heidolph overhead stirrer RZR-2000 (Kelheim, Germany) at 80 °C and 250 rpm for 30 min. Then, 0.04 mmol catalyst (DBTDL) was added and the stirring was carried out at 80 °C and 80 rpm for 2 h. The amount of free NCO in the prepolymer was monitored by dibutylamine titration. Once the desired free NCO content was obtained, the chain extender (BD) was added and stirred at 80 °C and 80 rpm for 5 min. Figure 1 shows the scheme of the synthesis of the TPUs.

Two TPUs were synthesized. TPU1 was synthesized with a mixture of 75 wt% PPG2000 and 25 wt% PPG450, and TPU2 was synthesized with a mixture of 50 wt% PPG2000 and 50 wt% PPG450. The same amount of 1,4 butanediol chain extender was added. The hard segments contents of TPU1 and TPU2 were calculated as the ratio of the amount of MDI by the amounts of MDI, polyols, and chain extender, and they were 21% and 28%, respectively. TPU1 was selected for its excellent tack and poor cohesion, whereas TPU2 was chosen for its good cohesion and poor tack [23].

### 2.3. Preparation of the Blends of TPUs in MEK Solutions

Methyl ethyl ketone (MEK) solutions of solid TPU1 and solid TPU2 were prepared for making the blends. Two different methods for obtaining the TPU1 + TPU2 blends were used.

In method A, 18 wt% solid TPU (15 g) was added to MEK (68.33 g) in a hermetically closed polypropylene container of 58 mm length and 68 mm diameter. The top of the container was sealed with parafilm (Parafilm, Bemis, Oshkosh, USA) and the mixture was magnetically stirred with a magnetic Teflon^®^ cylindrical stirrer of 20 mm length and 8 mm diameter in an IKA C-MAG HS 7 stirrer (IKA, Staufen, Germany) at 25 °C and 60 rpm for at least 30 min. Because of the solvent evaporation during the preparation of the solutions, additional MEK was added to adjust the solids content to 18 wt%. Afterwards, 60–80 wt% TPU1 solution in MEK and 20–40 wt% TPU2 solution in MEK were added in a hermetically closed polypropylene container of 58 mm length and 68 mm diameter and stirred with a magnetic Teflon^®^ cylindrical stirrer of 20 mm length and 8 mm diameter in an IKA C-MAG HS 7 stirrer (IKA, Staufen, Germany) at 25 °C and 60 rpm for at least 30 min. Figure 2 shows the scheme of the procedure employed to prepare the blends of TPU1 and TPU2 (TPU1 + TPU2) by using method A. The nomenclature of the blends consists of the amount of each TPU in wt% followed by the capital letters “TPU” and “/”, ending with the capital letter “A” between brackets. For example, 80TPU1/20TPU2 (A) corresponds to the blend made with 80 wt% TPU1 solution in MEK and 20 wt% TPU2 solution in MEK by using method A.

In method B, 80 wt% solid TPU1 and 20 wt% solid TPU2 were added together in a closed cylindrical polypropylene container (58 mm length and 68 mm diameter) and MEK was added to obtain a final solids content of 18 wt%. The top of the container was sealed with parafilm (Parafilm, Bemis, Oshkosh, USA). The mixture was magnetically stirred with a magnetic Teflon^®^ cylindrical stirrer of 20 mm length and 8 mm diameter in an IKA C-MAG HS 7 stirrer (IKA, Staufen, Germany) at 25 °C and 60 rpm for at least 60 min. The resulting blend was named 80TPU1/20TPU2 (B). Figure 2 shows the scheme of the procedure employed to obtain the 80TPU1/20TPU2 (B) blend.

MEK from the solutions of TPU1, TPU2, and TPU1 + TPU2 blends was removed for preparing solid films. The solid films were obtained by placing 4 g of TPU solution in MEK in an open cylindrical polypropylene mold of 30 mm height and 25 mm diameter, allowing the solvent removal at room temperature for 72 h.

### 2.4. Preparation of the TPU PSAs

PSAs consist of an adhesive supported on a thin substrate. TPU PSAs were prepared by placing the TPU solution in MEK on a 50 µm thick polyethylene terephthalate (PET) film; before applying the TPU solution, the PET film was wiped with MEK. The TPU solution was applied to the PET film with a pipette and spread by means of a metering rod of 400 μm. Then, the solvent was removed at room temperature for 24 h to obtain a dry TPU film on 30–40 µm thick PET film.

### 2.5. Experimental Techniques

#### 2.5.1. Solids Content

The solids contents of the TPUs were obtained by the difference in the weights before and after the evaporation of the solvent. Approximately 0.5 g TPU solution was placed and spread by means of a Pasteur pipette on aluminum foil, and the solvent was evaporated at 50 °C in an oven until a constant weight was obtained. Two replicates were tested and averaged for each TPU solution.

#### 2.5.2. Attenuated Total Reflection Infrared Spectroscopy (ATR-IR)

The ATR-IR spectra of TPU1, TPU2, and TPU1 + TPU2 blends were obtained in a Tensor 27 FT-IR spectrometer (Bruker Optik GmbH, Erlinger, Germany) by using a Golden Gate single reflection diamond ATR accessory. The angle of the incident beam was 45° and 64 scans were recorded in absorbance mode with a resolution of 4 cm^−1^ in the wavenumber range of 4000 to 400 cm^−1^.

#### 2.5.3. Differential Scanning Calorimetry (DSC)

The thermal and structural properties of TPU1, TPU2, and TPU1 + TPU2 blends were determined in a DSC Q100 calorimeter (TA Instruments, New Castle, DE, USA) under nitrogen atmosphere (flow: 50 mL/min). A total of 8–9 mg of TPU film was placed in a hermetically sealed aluminum pan and placed in the oven of the DSC equipment. For removing the thermal history, the TPU film was heated from −80 to 100 °C using a heating rate of 10 °C/min. Then, the TPU was cooled down to −80 °C using a cooling rate of 10 °C/min and, finally, the TPU was heated again from −80 to 150 °C using a heating rate of 10 °C/min. The glass transition temperature (*T*_g_) of TPU1, TPU2, and TPU1 + TPU2 blends were obtained from the second DSC heating run.

#### 2.5.4. Thermal Gravimetric Analysis (TGA)

The thermal degradation and the structure of TPU1, TPU2, and TPU1 + TPU2 blends were determined in a TGA Q500 equipment (TA Instruments, New Castle, USA) under nitrogen atmosphere (flow: 50 mL/min). A total of 8–9 mg of TPU was placed in platinum crucible and then heated from 35 to 800 °C using a heating rate of 10 °C/min.

#### 2.5.5. Plate-Plate Rheology

The rheological and viscoelastic properties of TPU1, TPU2, and TPU1 + TPU2 blends were assessed in a DHR-2 rheometer (TA Instruments, New Castle, DE, USA) using parallel plate-plate geometry. The gap selected was 0.40 mm, and 20 mm diameter stainless steel parallel plates were used. Temperature sweep experiments were carried out from −10 to 120 °C, a frequency of 1 Hz and a heating rate of 5 °C/min were used. Oscillatory frequency sweep experiments were also performed at 25 °C using 2.5% strain amplitude in the angular frequency range from 0.01 to 100 rad s^−1^. All rheological experiments were carried out in the region of linear viscoelasticity.

#### 2.5.6. Probe Tack

The probe tack of the TPU PSAs (TPU on PET films) was measured in the range of 15 to 50 °C with an interval of 5 °C in a TA.XT2i Texture Analyzer (Stable Micro Systems, Surrey, UK). The TPU PSAs were attached to square stainless steel 304 plates of 6 cm × 6 cm × 0.1 cm by means of double-sided tape (Miarco, Paterna, Spain). A flat end cylindrical stainless-steel probe of 3 mm diameter was used. The probe was brought into contact with the TPU PSA surface for 1 s under a load of 5 N. Then, the probe was pulled out at a constant rate of 10 mm/s and a stress–strain curve was obtained. The maximum of the stress–strain curve was taken at the tack of the TPU PSA. At least five replicates were carried out and averaged.

#### 2.5.7. Creep Test under Shear

Pieces of the TPU PSAs (TPU on PET films) were cut to obtain strips of 2.4 cm × 20 cm. On the other hand, rectangular pieces of polished stainless steel 304 of 77 cm × 51 cm × 1.5 cm were wiped with MEK to remove surface contaminants, allowing the solvent to evaporate for 15 min. Then, the TPU PSA strip was placed in the central area of the clean polished stainless steel 304 piece, an area of 2.4 cm × 2.4 cm was joined, and a rubber coated roller of 2 kg was passed 30 times over the joint. Afterwards, a piece of stainless steel was placed at the bottom of the TPU PSA strip, which was plied and fixed with a staple at a distance of 4 cm from the polished stainless-steel plate. The coupon was placed on the holder of a Shear-10 equipment (ChemInstruments, Fairfell, OH, USA) and 1 Kg weight was held at the bottom (Figure 3). The creep resistance at 25 °C, which is related to the cohesion, was obtained as the “holding time”, i.e., the time needed for the TPU PSA strip to fall down. Three replicates were tested for each TPU PSA and the results obtained were averaged.

#### 2.5.8. 180° Peel Test

The adhesion properties of the TPU PSAs (TPU on PET films) were determined by 180° peel tests of stainless steel 304/TPU PSA joints. TPU PSA strips of 30 mm × 180 mm × 0.50 mm were placed on a stainless-steel 304 plate of 30 mm × 150 mm × 1 mm and a 2 Kg rubber coated roller was passed 30 times over the joint. After 30 and 72 min of the joints formation, the 180° peel tests (Figure 4) were carried out in an Instron 4411 universal testing machine (Instron Ltd., Buckinghamshire, UK) using a pulling rate of 152 mm/min. A length of 7 cm of each joint was peeled, and the initial values of 180° peel strength were discarded. Five replicates were tested and averaged for each joint.

## 3. Results and Discussion

### 3.1. Influence of the Procedure for Preparing the Blends of TPUs on Their Properties

In a recent study [23], TPU1 PSA synthetized with 75 wt% PPG polyol with molecular weight 2000 Da (PPG2000) and 25 wt% PPG with molecular weight 450 Da (PPG450) showed high tack (752 kPa) but low cohesion at 25 °C, whereas TPU2 PSA synthetized with 50 wt% PPG2000 and 50 wt% PPG450 showed low tack (295 kPa) but high cohesion at 25 °C. For balancing the tack and the cohesion of the TPU PSAs, different blends of TPU1 and TPU2 were prepared. Two different procedures for preparing the blends were used (Figure 2): (i) Method A—different amounts of TPU1 solution in MEK and TPU2 solution in MEK were blended; (ii) Method B—80 wt% solid TPU1 and 20 wt% solid TPU2 were dissolved together in MEK. For determining the influence of the procedure for preparing the blends of the TPUs on their properties, the blend of 80 wt% TPU1 and 20 wt% TPU2 was selected.

The chemical structure of the blends of the TPUs was assessed by ATR-IR spectroscopy. Figure 5a shows the ATR-IR spectra of the blends prepared with the two procedures. Both blends showed the same chemical structure. The ATR-IR spectra show the bands of the PPG soft segments due to the asymmetric and symmetric C–H stretching of the hydrocarbon chains at 2971 and 2869 cm^−1^, CH_3_ and CH_2_ bands at 1373—scissor and rocking CH_3_ (sym), 1453—scissor and rocking CH_3_ + scissor and rocking CH_2_, and 927 cm^−1^—CH_2_ bending, and the strong band at 1084 cm^−1^ due to the asymmetric stretching of C−O−C. The bands corresponding to the hard segments (urethane groups) appeared at 3300 cm^−1^—symmetric and asymmetric N−H stretching, 1598 cm^−1^—in plane N−H bending, and 1727 cm^−1^—C=O stretching. Furthermore, the typical C=C stretching and bending in the benzene ring of MDI at 1412, 818, and 512 cm^−1^ were also observed.

The existence of hydrogen bond formation in polyurethanes by IR spectroscopy has been extensively studied in the existing literature [24,25]. The hydrogen bond interactions between the N–H and C=O groups of the urethane groups in the hard segments (associated urethanes) were assessed from the ATR-IR spectra of Figure 5a. The relative percentages of the free and associated urethane groups were assessed by curve fitting of the C=O band of the ATR-IR spectra of the blends. Figure 5b shows a typical example of the curve fitting of the carbonyl band of 80TPU1/20TPU2 (A) blend; the curve fitting was carried out by assuming a Gaussian distribution. The free urethane groups were fitted at 1727 cm^−1^ and the associated urethane groups were fitted at 1706–1705 cm^−1^. According to Table 2, the free urethane groups were dominant in the structure of both TPU1 + TPU2 blends, and similar contributions of the free and associated urethane groups were evidenced in both blends.

The structure of the blends was also determined by DSC. The DSC thermograms of the 80TPU1/20TPU2 (A) and 80TPU1/20TPU2 (B) blends are shown in Figure 6, and they exhibit the glass transition temperatures due to the soft segments at −30 °C and −32 °C, respectively. The small differences between the glass transition temperatures of the blends suggest slight differences in the degree of phase separation between the soft and the hard segments.

The viscoelastic properties of 80TPU1/20TPU2 (A) and 80TPU1/20TPU2 (B) blends were determined by plate-plate rheology (temperature sweep experiments). Figure 7a shows the variation in the storage modulus (*G*’) as a function of the temperature for the blends; at any temperature, the storage modulus was higher in the 80TPU1/20TPU2 (A) blend. On the other hand, a cross-over between the storage (*G*’) and the loss (*G*’’) moduli was found in both blends (Figure 7b,c). Figure 7b,c show that below 53 °C or 46 °C, respectively, the elastic rheological regime was dominant in the blends, whereas above 53 °C or 46 °C the viscous rheological regime was prevailing. The viscous behavior of the blends is mainly determined by their soft segments and, consequently, their content in the blend will determine the values of the temperature and the modulus at the cross-over of *G*’ and *G*’’. According to Figure 7b,c, and Table 3, the 80TPU1/20TPU2 (B) blend had a slightly lower cross-over temperature than the 80TPU1/20TPU2 (A) blend, likely due to a slightly different degree of phase separation; however, both blends showed similar moduli at the cross-over of *G*’ and *G*’’.

The properties of the TPU PSAs made with 80 wt% TPU1 + 20 wt% TPU2 blends on PET film were characterized by tack measurements at different temperatures, cohesion (holding time) at 25 °C, and 180° peel strength of stainless steel/TPU PSA joints at 25 °C.

Figure 8 shows the variation in the tack as a function of the temperature for 80TPU1/20TPU2 (A) and 80TPU1/20TPU2 (B) PSAs. The tack of both TPU PSAs was higher than 300 kPa and decreased by decreasing the temperature from 50 to 15 °C; furthermore, the tack values at any temperature were higher in the 80TPU1/20TPU2 (A) PSA, indicating that the procedure to prepare the blend determines its tack. The higher tack of 80TPU1/20TPU2 (A) PSA can be ascribed to its slightly higher degree of phase separation, and its higher storage modulus and *T*_cross-over_. On the other hand, both TPU PSAs showed good cohesion (i.e., high holding time) and acceptable 180° peel strength (Table 4), and the 80TPU1/20TPU2 (A) PSA had higher 180° peel strength and lower cohesion than 80TPU1/20TPU2 (B) PSA. However, the failed surfaces after the 180° peel test show a cohesive failure in the blend, which is not desirable. Therefore, TPU1 + TPU2 blends with different compositions were prepared and characterized.

### 3.2. Characterization of the TPU1 + TPU2 Blends

The main target of this study is the development of TPU PSAs with balanced adhesion and cohesion properties, and because the best balance between tack, 180° peel strength, and cohesion was obtained in 80TPU1/20TPU2 (A) PSA, the procedure selected for preparing other TPU1 + TPU2 blends was method A. Table 5 shows the composition of the TPU1 + TPU2 blends (TPU1 content between 60 and 80 wt%). The solids contents of TPU1 and TPU2 were 18.8 wt% and 19.9 wt%, respectively, and the contents of the TPU1 + TPU2 blends were 17.8–19.3 wt%.

The chemical structures of the TPU1 + TPU2 blends were characterized by ATR-IR spectroscopy (Figure 9a). The absorption bands of the hard segments can be distinguished at 3310–3255 (N–H stretching), 1727–1726 (C=O stretching of urethane), 1533–1532 (C–N stretching), and 1598 cm^−1^ (N–H bending in plane), whereas the absorption bands of the soft segments were located at 2972–2971 and 2869–2868 cm^−1^ (asymmetric and symmetric C–H stretching, respectively), and at 927, 1084–1017, 1373, and 1454–1453 cm^−1^ (–C–O–C– group of PPG polyol).

In order to assess the contributions of the free and hydrogen-bonded urethane groups in TPU1, TPU2, and TPU1 + TPU2 blends, the carbonyl region of the ATR-IR spectra was curve fitted (Figure 9b). According to Table 6, the free urethane (fitted at 1727–1726 cm^−1^) was dominant in TPU1, whereas the hydrogen-bonded urethane (fitted at 1706–1704 cm^−1^) was the major contribution in TPU2. Interestingly, the TPU1 + TPU2 blends showed somewhat similar contributions of the free and hydrogen-bonded urethane groups, indicating that they have almost similar degrees of phase separation.

The structural changes and the degree of phase separation in TPU1, TPU2, and TPU1 + TPU2 blends were determined by DSC. Figure 10 shows the DSC thermograms (second heating run) of TPU1, TPU2, and TPU1 + TPU2 blends. At low temperature, all TPUs showed the glass transition temperature of the soft segments (*T*_g_), the lowest *T*_g_ corresponds to TPU1 (−36 °C) and the highest to TPU2 (−16 °C). Interestingly, all TPU1 + TPU2 blends had similar *T*_g_ values (near −30 °C), irrespective of the composition of the blend, indicating that the structure of the soft segments was similar in all blends and it was somewhat similar to the one of TPU1. In agreement with previous findings [23], the increase in the hard segment content in the TPUs increased their *T*_g_ values and the extent of mixing of the hard and soft segments, i.e., the degree of microphase separation, decreased. Therefore, similar structure of the soft segments and analogous degree of microphase separation of the soft and hard segments was obtained in the TPU1 + TPU2 blends, both were different than the ones in the parent TPUs, in agreement with the evidence provided by ATR-IR spectroscopy.

The thermal stabilities and the structure of TPU1, TPU2, and TPU1 + TPU2 blends were analyzed by TGA. Figure 11a shows that TPU1 had the highest thermal stability and TPU2 the lowest. The thermal stabilities of TPU1, TPU2, and TPU1 + TPU2 blends were quantified by the values of the temperatures at which 5 (*T*_5%_) and 50 wt% (*T*_50%_) were lost. According to Table 7, the values of *T*_5%_ and *T*_50%_ of the blends decrease by increasing their TPU2 content. On the other hand, two main thermal decompositions can be distinguished in the TPUs and their blends (Figure 11b), one at 308–317 °C due to the urethane hard domains and another at 363–373 °C due to the soft domains [26]. The weight loss of the hard domains increased and the one of the soft domains decreased by increasing the amount of TPU2 in the blends, and the temperatures of the thermal decompositions were similar in 80TPU1/20TPU2 and 60TPU1/40TPU2; however, the temperatures of decomposition of the hard and soft domains were higher in 70TPU1/30TPU2 (Table 7).

The structure of the TPUs affects their viscoelastic properties. The viscoelastic properties of TPU1, TPU2, and TPU1 + TPU2 blends were determined by plate-plate rheology experiments. Figure 12a,b show the variation in the storage modulus (*G*’) and the loss modulus (*G*’’) as a function of the temperature for TPU1, TPU2, and TPU1 + TPU2 blends. The storage and the loss moduli decreased by increasing the temperature, more noticeably in TPU1 than in TPU2, and the storage and loss moduli of the TPU1 + TPU2 blends were intermediate between the ones of TPU1 and TPU2. For temperatures below 20 °C, the storage and loss moduli of the TPU1 + TPU2 blends were similar, but above 20 °C the moduli were higher in the blends with higher content of TPU2. All TPUs and TPU1 + TPU2 blends showed a cross-over of the storage (*G*’) and loss (*G*’’) moduli (Figure 7b) and the values of the temperatures and the moduli at the cross-over are given in Table 8. The temperature at the cross-over of *G*’ and *G*’’ was lower in TPU1 than in TPU2 because of the lower content of PPG450 polyol, and the temperatures at the cross-over in the blends increased by increasing their TPU2 content. Interestingly, the moduli at the cross-over were higher in TPU1 and TPU2 than in the blends, this can be related to their lower degree of phase separation.

The unexpected particular structures of the TPU1 + TPU2 blends with respect to the ones of the parent TPUs leading to lower degree of phase separation should derive from the structural changes produced when the solvent (MEK) in the solutions is removed. It has been shown [27,28,29] that the interactions by hydrogen bonds between the polymer chains in the TPUs can be reversibly destroyed by increasing the temperature or by adding organic solvents (particularly ketones). In this study, the solid TPU1 and solid TPU2 were dissolved in MEK, which caused the rupture of the hydrogen bonds between the hard segments (Figure 13a,b). The structures of TPU1 and TPU2 were re-formed upon MEK removal, i.e., the hydrogen bonds between the hard segments were created. However, when the MEK solutions of TPU1 and TPU2 were mixed and the solvent was removed, the structure was different because the interactions between the hard domains were more complex and a higher number of hydrogen bonds were formed (Figure 13c), this led to a lower degree of phase separation between the soft and the hard domains. As a consequence, the structures of the TPU1 + TPU2 blends were different than the ones in TPU1 and TPU2.

The experimental results shown above indicate that the most efficient TPU1 + TPU2 blends were obtained by adding 20–30 wt% TPU2, likely due to easy mobility of the polymeric chains of TPU2 during MEK removal. Because the number of hydrogen bond interactions in the TPUs are tightly related to their cohesion, higher cohesion in the TPU1 + TPU2 blends than in TPU1 can be anticipated; however, at the same time the mobility of the polymeric chains of TPU2 should be reduced, so a decrease in tack can be expected. The properties of the TPU PSAs made with TPU1, TPU2, and TPU1 + TPU2 blends are studied in the next section.

### 3.3. Characterization of the TPU1 + TPU2 PSAs

The performance of the PSAs is tightly related to their viscoelastic properties. The TPU PSAs were made by placing the MEK solutions of TPU1, TPU2, and TPU1 + TPU2 blends on PET film. The most PSAs are used at ambient temperature, so the viscoelastic properties of the TPU PSAs were studied at 25 °C by oscillatory frequency sweep plate-plate rheology experiments. The storage modulus (*G*’) at low frequency of the TPU PSA is related to its tack and shear resistance, whereas at higher frequencies the *G*’ is associated with its peel strength. An excellent PSA must have low *G*’ value at high frequency (high shear and easy peel) and high *G*’ value at low frequency (good resistance to creep). Figure 14a shows that the TPU1 PSA had low *G*’ values at low frequencies, anticipating poor cohesion, but the TPU2 PSA had high *G*’ values in all range of frequencies, anticipating high cohesion; the *G*’ values of the TPU1 + TPU2 PSAs showed reasonable values and they must have good cohesion. On the other hand, all TPU PSAs had high *G*’ values at high frequencies, anticipating that they should have easy peel. Furthermore, the variation in the loss modulus (*G*’’) as a function of the frequency (Figure 14b) shows the same trend as the ones of the storage modulus (Figure 14a), but the differences in the loss moduli are less marked.

Table 9 compiles the values of *G*’ at 0.1 and 100 rad/s. TPU1 PSA had low *G*’ value at 0.1 rad/s and high *G*’ (0.1 rad/s)/*G*’ (100 rad/s) value, which is typical of PSAs with high tack [30]. On the contrary, the *G*’ value at 0.1 rad/s was high and the *G*’ (0.1 rad/s)/*G*’ (100 rad/s) ratio was low in TPU2 PSA, and its tack should be low. The *G*’ value at 0.1 rad/s of the TPU1 + TPU2 PSAs increased and their *G*’ (0.1 rad/s)/*G*’ (100 rad/s) ratios decreased by increasing their TPU2 content; according to the values in Table 9, 60TPU1/40TPU2 PSA should have low tack, easy peel, and good creep resistance, but 70TPU1/30TPU2 PSA and 80TPU1/20TPU2 PSA should have a good balance of tack, cohesion, and peel.

Chang’s viscoelastic window is a simple tool to determine the balance of the viscoelastic properties of the PSAs [31]. Chang’s viscoelastic windows at 25 °C of TPU1, TPU2, and TPU1 + TPU2 PSAs are given in Figure 15. The differences between the storage (*G*’) and loss (*G*’’) moduli were lower in the TPU1 + TPU2 PSAs with respect to TPU1 PSA and TPU2 PSA, thus anticipating a better compromise between tack and cohesion. All TPU1 + TPU2 PSAs had *G*’ values at 0.01 rad/s lower than 3·10^5^ Pa, anticipating an efficient contact with the substrate, and they followed the Dalhquist criterion—Dahlquist suggested that the tack of the PSAs requires a storage modulus value lower than 3·10^5^ Pa at 25 °C and 1Hz [32]. Furthermore, the TPU1 + TPU2 PSAs are good general-purpose PSAs (i.e., they are located in the center of the Chang’s viscoelastic window).

Figure 16 shows the variation of the tack of the TPU PSAs as a function of the temperature. TPU1 PSA showed the highest tack at any temperature and the tack was maintained between 15 and 30 °C, and an increase in production above 30 °C. A similar trend and slightly lower tack values were obtained in 80TPU1/20TPU2 PSA, whereas the tack was lower and similar between 15 and 40 °C in 70TPU1/30TPU2 PSA. The lowest tack values corresponded to TPU2 PSA and the maximum tack was obtained at 25–30 °C. At 25 °C, the tack ranged between 391 and 634 kPa and was somewhat similar in all TPU1 + TPU2 PSAs with TPU1 content of 70 wt% or lower; however, at 15 and 30 °C the tack of the TPU PSAs decreased by increasing their content in TPU2, which can be related to the decrease in the soft segments content. In summary, the tack of the TPU PSAs can be designed by changing their soft segments contents and its variation with the temperature is related to their viscoelastic properties.

Apart from an adequate tack, the PSAs must possess an adequate peel strength and sufficient cohesion. Table 10 shows the 180° peel strength values of the stainless steel 304/TPU PSA film joints measured after 30 min of joint formation. The 180° peel strengths of the joints made with TPU1 PSA and TPU2 PSA were low and their loci of failure were different, i.e., the joint made with TPU1 PSA failed cohesively in the adhesive, which is not acceptable in PSAs, whereas an adhesion failure to the stainless-steel substrate was obtained in the joint made with TPU2 PSA. The loci of failure of the joints made with TPU1 and TPU2 PSAs are related to their cohesion or holding time, which was quite low in TPU1 PSA and quite high in TPU2 PSA. Interestingly, the 180° peel strength values of the joints made with all TPU1 + TPU2 PSAs were higher than the ones of TPU1 and TPU2 PSAs and the highest 180° peel strengths corresponded to the joints made with 80TPU1/20TPU2 and 70TPU1/30TPU2 PSAs. Because the holding time was higher in 70TPU1/30TPU2 PSA, the joints made with this TPU PSA showed an adhesion failure to the stainless-steel. In summary, the 70TPU1/30TPU2 PSA showed an excellent balance of tack, peel, and cohesion for being used as general purpose pressure sensitive adhesive. This balance of properties is due to the adequate soft segments content and the formation of new hydrogen bonded urethane interactions, which causes a lower degree of phase separation.

In order to demonstrate the potential of the novel TPU PSAs, their pressure sensitive adhesive properties were compared to the ones of a commercial polyurethane pressure sensitive adhesive (SPUR PSA 3.0, Momentive, UK) of unknown formulation supported on PET film. For the commercial polyurethane PSA, the tack at 25 °C was 504 ± 22 kPa, the 180° peel strength of stainless steel/commercial polyurethane PSA joints was 0.41 ± 0.03 N/m, and the holding time was 359 ± 3 min. Therefore, the TPU PSAs developed in this study show comparable tack to commercial polyurethane PSA but higher 180° peel strength and higher creep resistance.

## 4. Conclusions

Pressure sensitive adhesives with balanced adhesion and cohesion properties were prepared by blending thermoplastic polyurethanes with different properties. The procedure for preparing the blends of the TPUs determined their different viscoelastic properties, and the properties of the TPU PSAs as well, the blending of separate MEK solutions of the two TPUs imparted higher tack and 180° peel strength, and adequate cohesion.

The TPU1 + TPU2 blends showed somewhat similar contributions of the free and hydrogen-bonded urethane groups, indicating that they had almost similar degrees of phase separation, which was lower than in the parent TPUs. All TPU1 + TPU2 blends showed the glass transition temperature of the soft segments at about −30 °C, which were similar irrespective of the composition of the blend, confirming that they showed a similar degree of microphase separation. Furthermore, two main thermal decompositions at 308–317 °C due to the urethane hard domains and another at 363–373 °C due to the soft domains could be distinguished in the TPU1 + TPU2 blends, the weight loss of the hard domains increased and the one of the soft domains decreased by increasing the amount of TPU2 in the blends. The storage moduli of the TPU1 + TPU2 blends were intermediate between the ones of TPU1 and TPU2 and they were similar for temperatures lower than 20 °C, and the moduli at the cross-over were lower than in the parent TPUs, which can be related to their lower degree of phase separation.

The improved properties of the TPU1 + TPU2 blends derived from the removal of the hydrogen bonds between the hard segments in the MEK solutions of TPU1 and TPU2 that were re-formed upon MEK removal, producing a different structure because the interactions between the hard domains were more complex and a higher number of hydrogen bonds were formed, which led to a lower degree of phase separation between the soft and the hard domains. The most efficient TPU1 + TPU2 blends were obtained by adding 20–30 wt% TPU2, likely due to the easy mobility of the polymeric chains of TPU2 during MEK removal. As a consequence, the properties of the TPU1 + TPU2 PSAs were improved because good tack, high 180° peel strength, and sufficient cohesion were obtained, particularly in 70TPU1/30TPU2 PSA. Therefore, the novel TPU PSAs can be used for manufacturing labels and tapes for medical and automotive applications in which tack can be maintained in a wide range of temperatures without sacrificing the cohesion and the peel strength.

## Figures and Tables

**Figure 1 polymers-11-01608-f001:**
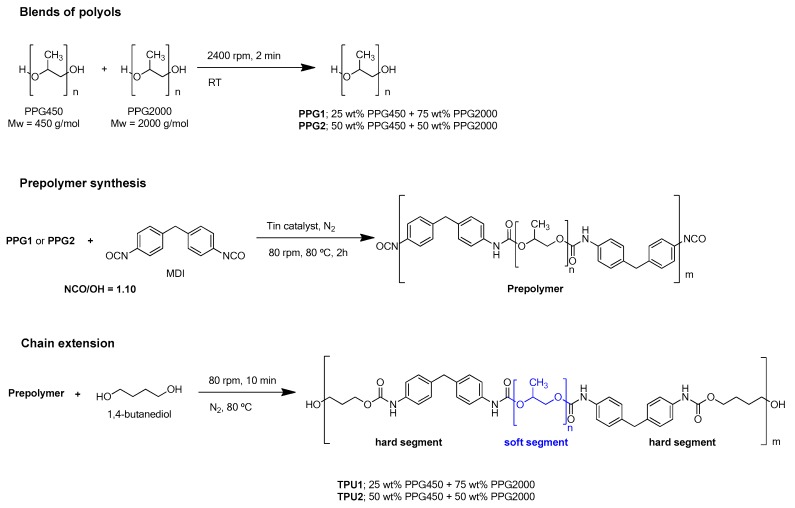
Scheme of the synthesis of the thermoplastic polyurethanes (TPUs).

**Figure 2 polymers-11-01608-f002:**
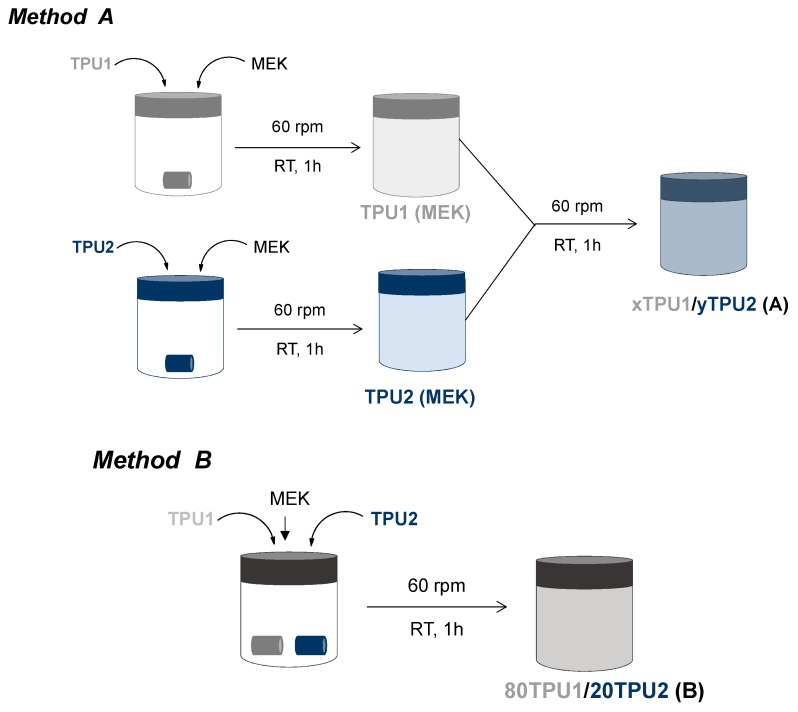
Schemes of the procedures used to prepare the blends of TPU1 and TPU2 in methyl ethyl ketone (MEK) solutions.

**Figure 3 polymers-11-01608-f003:**
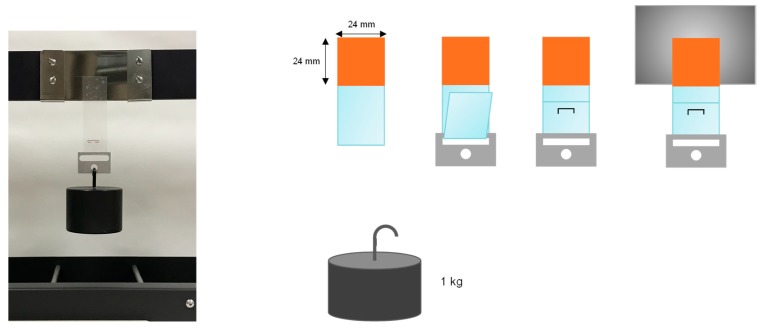
Scheme of the manufacturing of the coupons and the creep test of TPU PSA.

**Figure 4 polymers-11-01608-f004:**
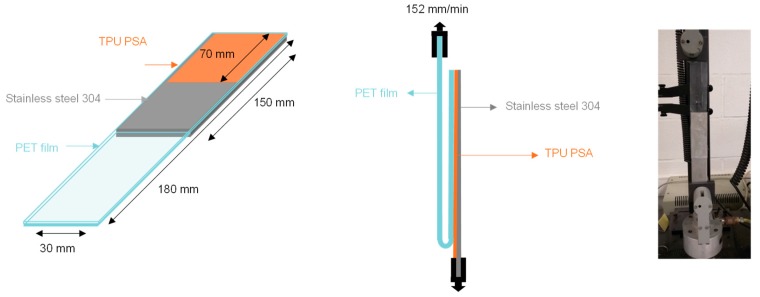
Scheme of the 180° peel strength test of a stainless-steel 304/TPU PSA joint.

**Figure 5 polymers-11-01608-f005:**
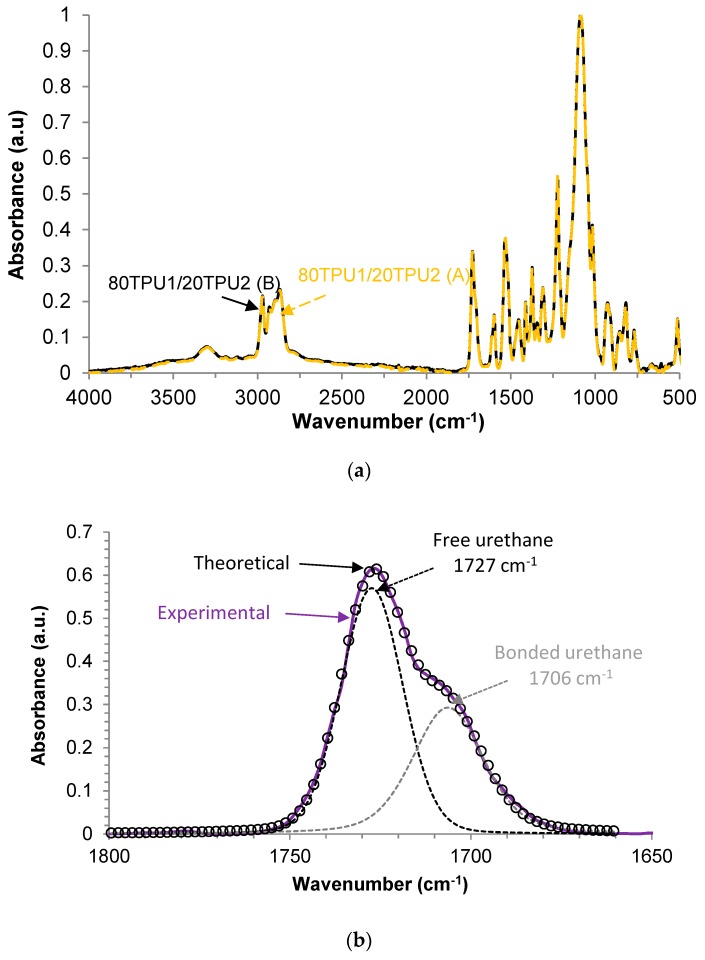
(**a**) ATR-IR spectra of 80TPU1/20TPU2 (A) and 80TPU1/20TPU2 (B) blends. (**b**) Curve fitting of the carbonyl region (1650–1800 cm^−1^) of the ATR-IR spectrum of the 80TPU1/20TPU2 (A) blend.

**Figure 6 polymers-11-01608-f006:**
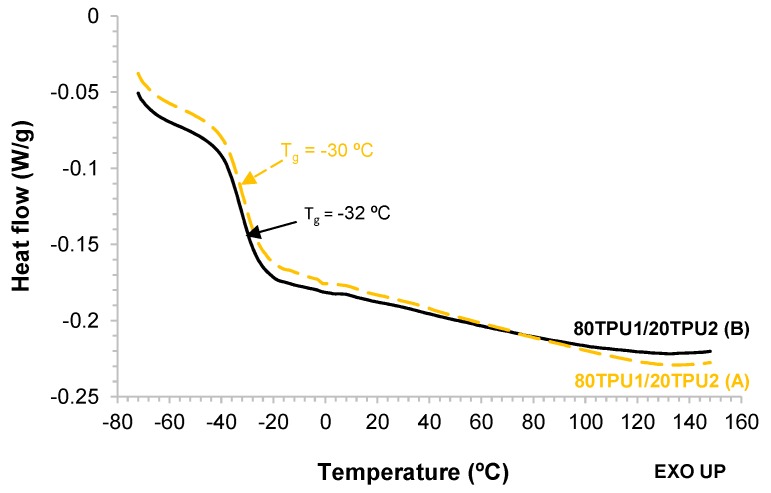
DSC thermograms of 80TPU1/20TPU2 (**A**) and 80TPU1/20TPU2 (**B**) blends. Second heating run.

**Figure 7 polymers-11-01608-f007:**
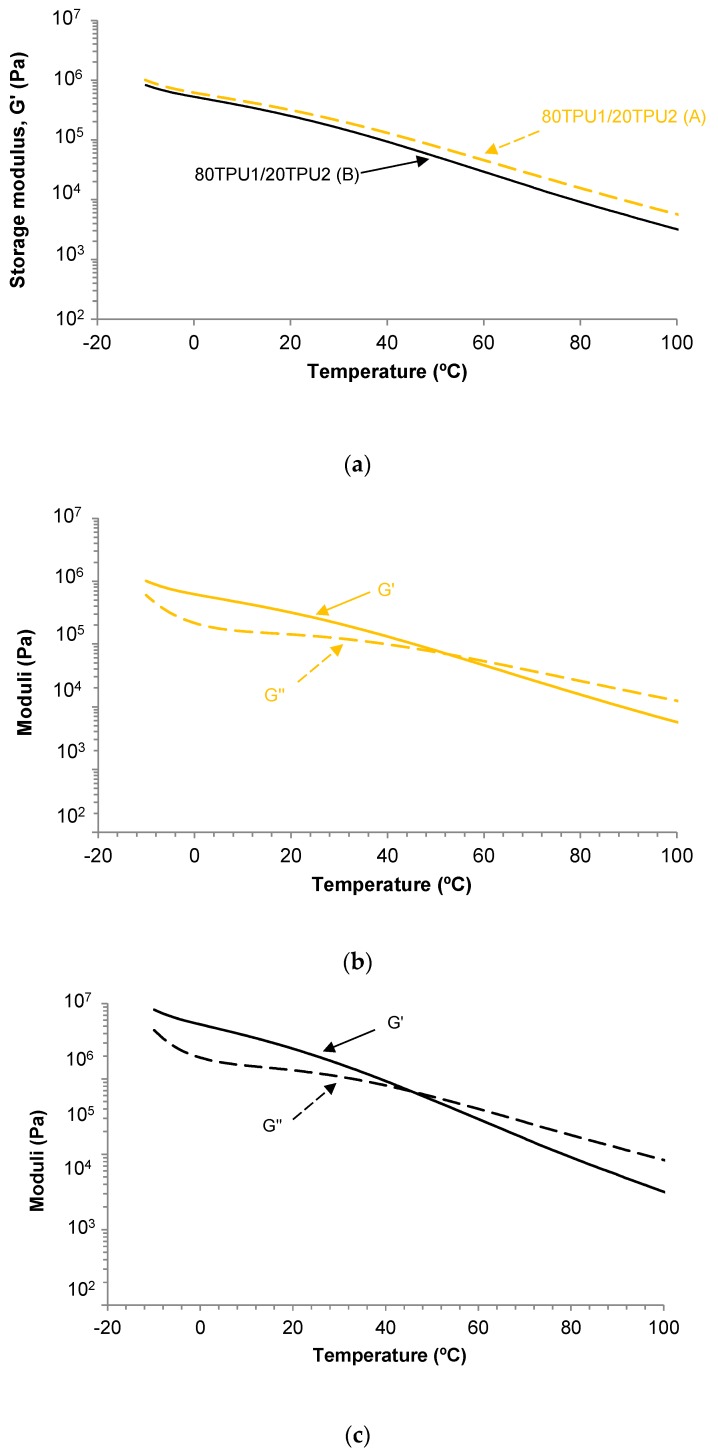
(**a**) Variation in the storage modulus (*G*’) as a function of the temperature for 80TPU1/20TPU2 (A) and 80TPU1/20TPU2 (B) blends. (**b**) Variation in the storage (*G*’) and loss (*G*’’) moduli as a function of the temperature for the 80TPU1/20TPU2 (A) blend. (**c**) Variation in the storage (*G*’) and loss (*G*’’) moduli as a function of the temperature for 80TPU1/20TPU2 (B) blend. Plate-plate rheology experiments.

**Figure 8 polymers-11-01608-f008:**
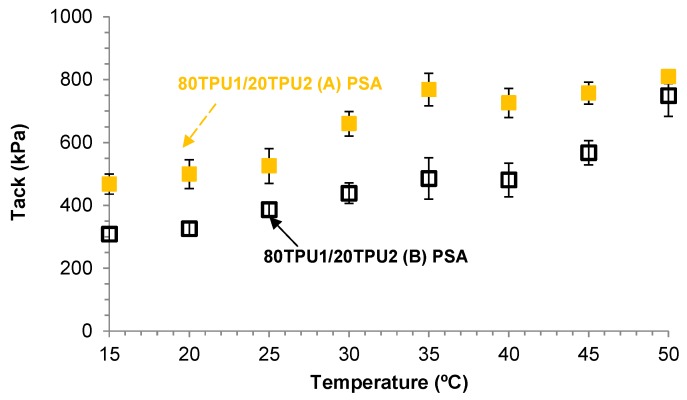
Variation in the tack of the 80TPU1/20TPU2 (**A**) PSA and 80TPU1/20TPU2 (**B**) PSA as a function of the temperature.

**Figure 9 polymers-11-01608-f009:**
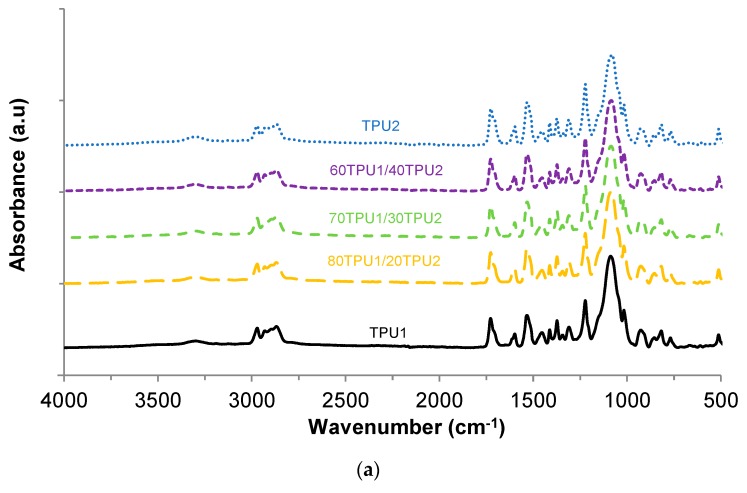
(**a**) ATR-IR spectra of TPU1, TPU2, and TPU1 + TPU2 blends. (**b**) Carbonyl region (1650–1800 cm^−1^) of the ATR-IR spectra of TPU1, TPU2, and TPU1 + TPU2 blends.

**Figure 10 polymers-11-01608-f010:**
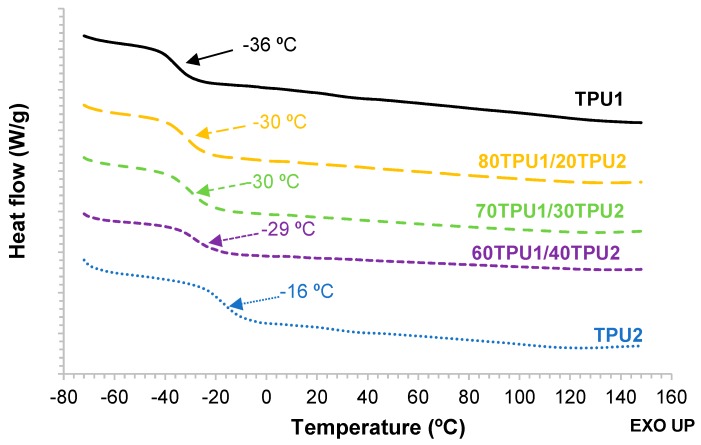
DSC thermograms of TPU1, TPU2, and TPU1 + TPU2 blends. Second DSC heating run.

**Figure 11 polymers-11-01608-f011:**
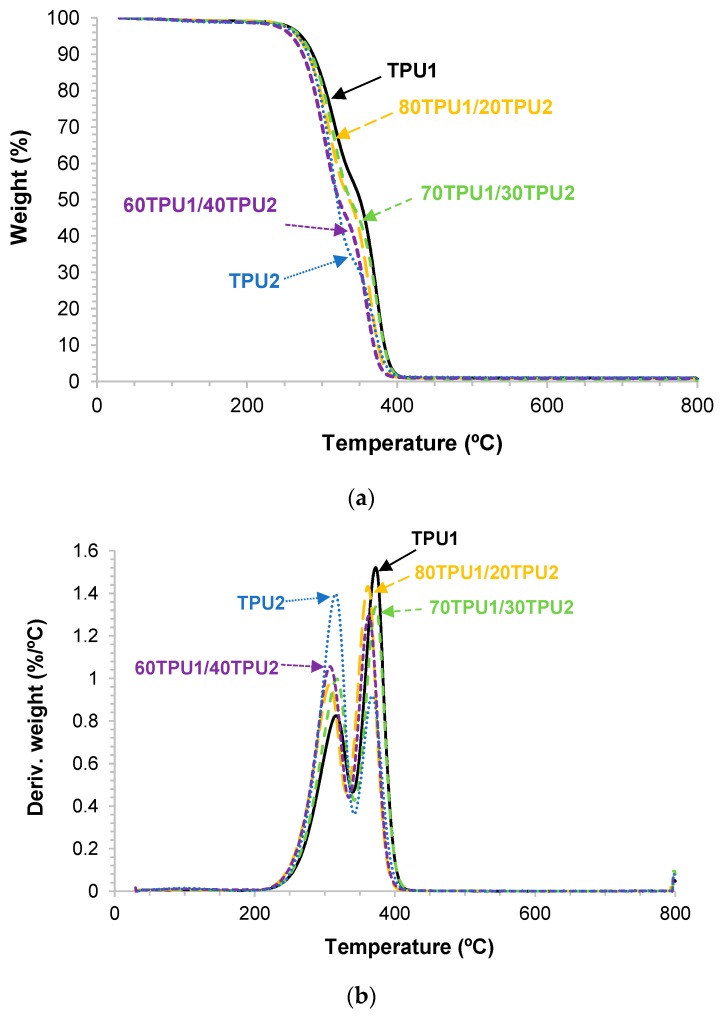
Variation in the (**a**) weight and (**b**) derivative of the weight as a function of the temperature for TPU1, TPU2, and TPU1 + TPU2 blends. TGA experiments.

**Figure 12 polymers-11-01608-f012:**
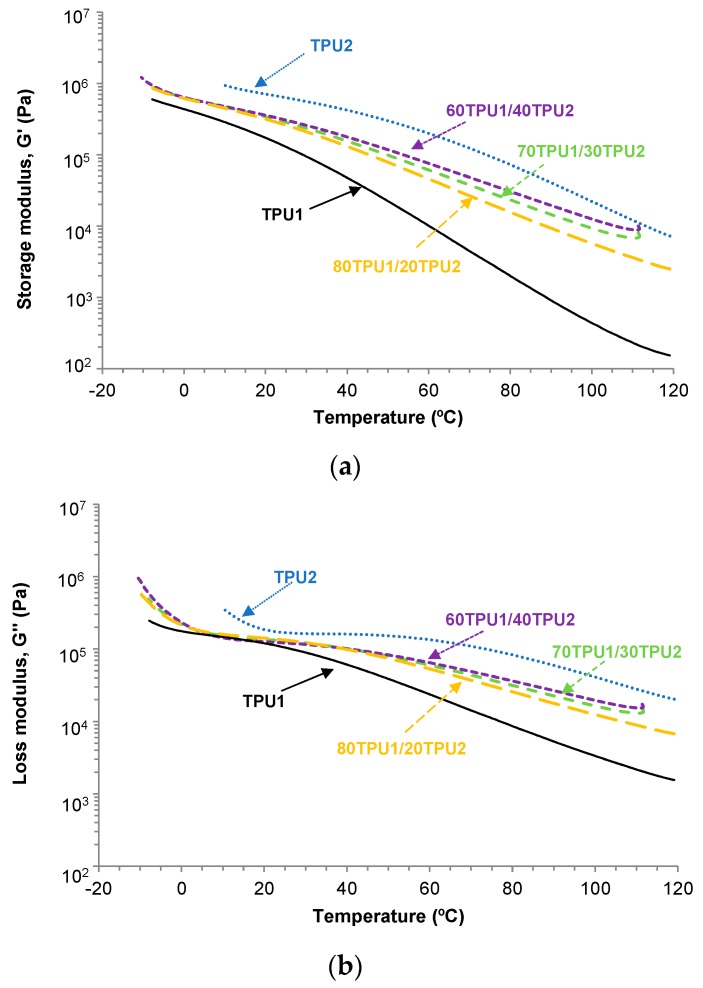
Variation in the (**a**) storage modulus (*G*’) and (**b**) loss modulus (*G*’’) as a function of the temperature for TPU1, TPU2, and TPU1 + TPU2 blends. Plate-plate rheology experiments. Temperature sweep.

**Figure 13 polymers-11-01608-f013:**
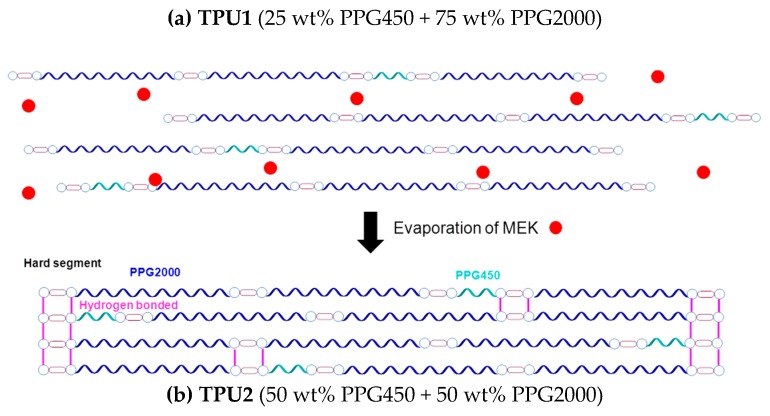
Scheme of the structure of TPU1, TPU2, and 70TPU1/30TPU2 solutions in MEK and after solvent removal.

**Figure 14 polymers-11-01608-f014:**
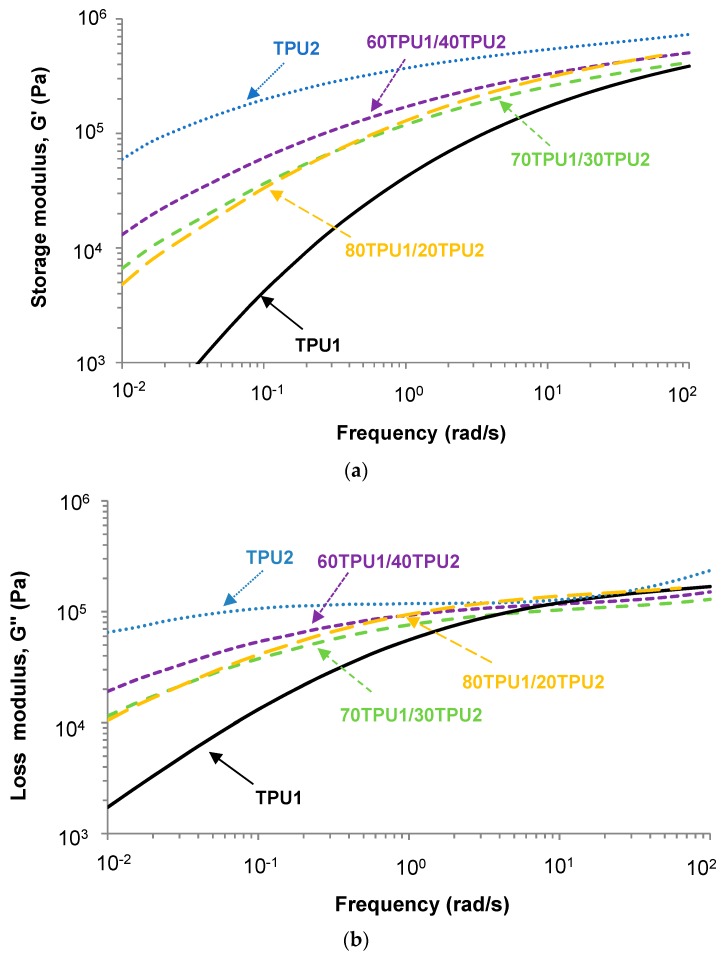
Variation in the (**a**) storage modulus (*G*’) and (**b**) loss modulus (*G*’’) at 25 °C as a function of the frequency of TPU1, TPU2, and TPU1 + TPU2 blends. Plate-plate rheology experiments. Frequency sweep.

**Figure 15 polymers-11-01608-f015:**
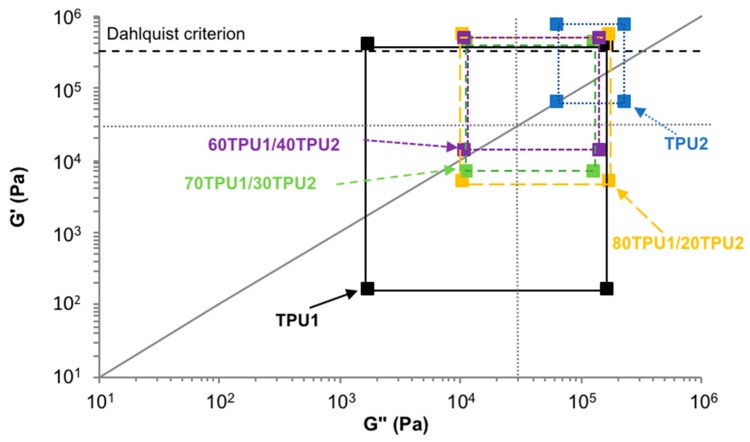
Chang’s viscoelastic windows at 25 °C of TPU1, TPU2, and TPU1 + TPU2 blends. Dotted lines indicate the four regions of Chang’s viscoelastic window. Solid line corresponds to *G*’ = *G*’’ (tan delta = 1). Dashed line indicates the Dahlquist criterion.

**Figure 16 polymers-11-01608-f016:**
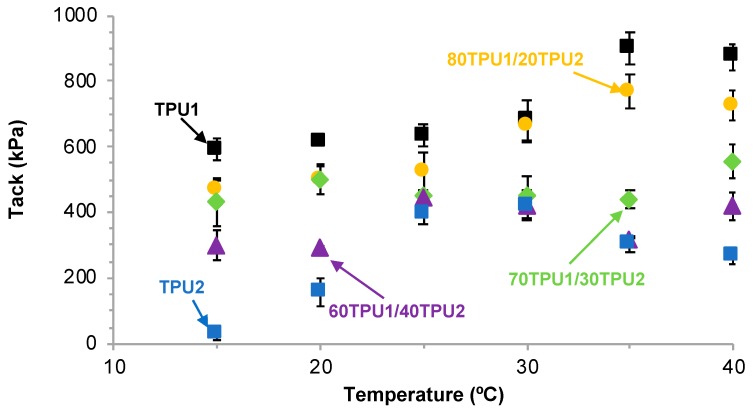
Variation of the tack of TPU1, TPU2, and TPU1 + TPU2 PSAs as a function of the temperature.

**Table 1 polymers-11-01608-t001:** Main properties of typical polymers used in pressure sensitive adhesives.

	Natural and Synthetic Rubber	Acrylic	Silicone	Polyurethane
Tack	High	Low to high	Low to high	Low
Peel strength	High	Medium to high	Medium	Low to medium
Cohesive strength	High	Low to high	High	High
Oxidative resistance	Poor	Good	Excellent	Excellent
Solvent resistance	Fair	High	Excellent	Excellent
Low skin sensibility	Poor to good	Good	Excellent	Good
Low skin trauma	Poor	Poor	Excellent	Good
Repositionability	Poor	Poor	Excellent	Fair
Cost	Low	Medium	High	Medium

**Table 2 polymers-11-01608-t002:** Relative contribution of the free and hydrogen bonded urethane groups of 80TPU1/20TPU2 (A) and 80TPU1/20TPU2 (B) blends. Curve fitting of the C=O band of the ATR-IR spectra.

	Relative Contribution of Species (%)	
Wavenumber (cm^−1^)	80TPU1/20TPU2 (A)	80TPU1/20TPU2 (B)
1727 cm^−1^ (Free urethane)	61	62
1706–1705 cm^−1^ (H-bonded urethane)	39	38

**Table 3 polymers-11-01608-t003:** Values of the temperature (*T*_cross-over_) and the modulus (*G*_cross-over_) at the cross-over of the storage and loss moduli of 80TPU1/20TPU2 (A) and 80TPU1/20TPU2 (B) blends. Plate-plate rheology experiments.

Blend	*T*_cross-over_ (°C)	*G*_cross-over_ (Pa)
80TPU1/20TPU2 (A)	53	6.8·10^4^
80TPU1/20TPU2 (B)	46	6.8·10^4^

**Table 4 polymers-11-01608-t004:** Holding time at 25 °C and 180° peel strength at 25 °C of stainless steel/TPU PSA joints.

PSA	Holding Time (min)	180° Peel Strength After 30 min (kN/m)	180° Peel Strength After 72 h (kN/m)
80TPU1/20TPU2 (A) PSA	442 ± 134	1.29 ± 0.06 (CA) ^a^	1.61 ± 0.07 (CA) ^a^
80TPU1/20TPU2 (B) PSA	845 ± 75	0.95 ± 0.03 (CA) ^a^	1.07 ± 0.02 (CA) ^a^

^a^ Locus of failure: CA, cohesive failure of the blend.

**Table 5 polymers-11-01608-t005:** Nomenclature and composition of the TPU1 + TPU2 blends (method A).

Sample Code	TPU1 (wt%)	TPU2 (wt%)	Solids Content (wt%)
TPU1	100	-	18.8 ± 1.7
80TPU1/20TPU2	80	20	17.9 ± 0.6
70TPU1/30TPU2	70	30	17.8 ± 0.7
60TPU1/40TPU2	60	40	19.3 ± 1.5
TPU2	-	100	19.9 ± 1.9

**Table 6 polymers-11-01608-t006:** Relative contributions of the free and hydrogen-bonded urethane groups of TPU1, TPU2, and TPU1 + TPU2 blends. Curve fitting of the C=O region of the ATR-IR spectra.

	Relative Contribution of Species (%)				
Wavenumber (cm^−1^)	TPU1	80TPU1/20TPU2	70TPU1/30TPU2	60TPU1/40TPU2	TPU2
1726–1727 cm^−1^ (Free urethane)	66	61	59	58	46
1706–1704 cm^−1^ (H-bonded urethane)	34	39	41	42	54

**Table 7 polymers-11-01608-t007:** Temperatures at which 5 wt% (*T*_5%_) and 50 wt% (*T*_50%_) were lost, and temperatures and weight losses of the two thermal decompositions of TPU1, TPU2, and TPU1 + TPU2 blends. TGA experiments.

Sample Code.	*T*_5%_ (°C)	*T*_50%_ (°C)	1st Decomposition		2nd Decomposition		Residue (wt%)
			T_1_ (°C)	Weight Loss_1_ (%)	T_2_ (°C)	Weight Loss_2_ (%)	
TPU1	272	351	309	44	373	54	2
80TPU1/20TPU2	267	336	309	48	365	51	1
70TPU1/30TPU2	271	338	317	52	373	48	0
60TPU1/40TPU2	263	326	308	53	363	45	2
TPU2	267	320	315	66	368	32	2

**Table 8 polymers-11-01608-t008:** Values of temperature (*T*_cross-over_) and modulus (*G*_cross-over_) at the cross-over of the storage and loss moduli of TPU1, TPU2, and TPU1 + TPU2 blends. Plate-plate rheology experiments.

Sample Code	*T*_cross-over_ (°C)	*G*_cross-over_ (Pa)
TPU1	32	8.3·10^4^
80TPU1/20TPU2	53	6.8·10^4^
70TPU1/30TPU2	61	5.7·10^4^
60TPU1/40TPU2	69	5.0·10^4^
TPU2	75	9.6·10^4^

**Table 9 polymers-11-01608-t009:** Values of the storage moduli (*G*’) at 25 °C and different frequencies for TPU PSAs. Frequency sweep plate-plate rheology experiments.

Sample Code	*G*’ (kPa)-0.1 rad/s	*G*’ (kPa)-100 rad/s	*G*’ (0.1 rad/s)/*G*’ (100 rad/s)
TPU 1	0.42	38.68	92.7
80TPU1/20TPU2	3.33	52.50	15.8
70TPU1/30TPU2	3.64	41.66	11.4
60TPU1/40TPU2	6.13	50.63	8.3
TPU2	10.68	23.55	2.2

**Table 10 polymers-11-01608-t010:** Tack values and holding times at 25 °C of TPU1, TPU2, and TPU1 + TPU2 PSAs, and 180° peel strength values at 25 °C of stainless steel/TPU PSA joints.

Sample Code	Tack at 25 °C (kPa)	180° Peel Strength (kN/m) ^a^	Holding Time at 25 °C (min)
TPU 1	634 ± 33	0.35 ± 0.04 (CA)	152 ± 46
80TPU1/20TPU2	525 ± 55	1.29 ± 0.06 (CA)	442 ± 134
70TPU1/30TPU2	450 ± 5	1.43 ± 0.25 (A)	847 ± 55
60TPU1/40TPU2	440 ± 21	0.85 ± 0.08 (A)	2115 ± 128
TPU2	391 ± 30	0.22 ± 0.04 (A)	4211 ± 10

^a^ Locus of failure: CA, cohesive failure of the blend; A, adhesion failure.

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
