# Peer review of "Balanced Viscoelastic Properties of Pressure Sensitive Adhesives Made with Thermoplastic Polyurethanes Blends"

_polymers, 2019, doi:10.3390/polym11101608_

Round 1
Reviewer 1 Report
The Authors performed interesting study how to tune the properties of the TPU-based materials. The idea as well as investigation route is very interesting, however there are some task those need to be performed as well as some remarks need to be responded adequately prior to publication. Please see below.
In my eyes more than 50% of references are more than 10 years old. THerefore the introduction section need to be improved by means of up to date literature references and thus the state of the art in field need to be highlighted by more references. The characterization of the fabricated TPUs are not properly characterized. NMR investigation is very important to se the real composition of the individual TPUs and also the content of 1,4-butanediol is not clear in the system and have the impact on the properties fo the materials. The XRD study is useless. Since the DSC does not show any melting peak, soever discussion on crystalline phase is worthless. Please delete this part from the manuscirpt. In the DSC (Fig. 6 and Fig. 9) there are highlighted two peaks and selected as peaks for polyols and soft segments. Two glass transitions need to be confirmed also by DMTA measurement using Rotational rheometer. Tan delta dependence on the temperature have to show two individual peaks. Otherwise, it need to be replaced. As asked in the former question, the DMTA from at least -60°C up to 160°C need to be performed nad Tan delta need to be plotted against Temperature. INto the the FIg. 8b, the blend type B need to be added as well as tan delta. In the table 10 I would suggest to change the name Blend to sample code. Blend is misleading. Similarly as for Fig.8 also Fig. 12 need to be enlarged to -60°C to 160°C and tand delta is misssing. In Fig.13 the G'' or tan delta is missing, the storage modulus is just one part and cannot be disccussed without lossses. Please add this and provide corresponding discussion. Generally, there are missing the references to the statements in results and discussion parts. Please add those.Author Response
See attached file

Reviewer 2 Report
In this manuscript, authors made pressure-sensitive adhesives with blends of thermoplastic polyurethanes with satisfactory tack, cohesion, and adhesion. They used two simple blending techniques to prepare pressure-sensitive adhesives. They characterized TPUs by ATR-IR spectroscopy, differential scanning calorimetry, X-ray diffraction, thermal gravimetric analysis and plate-plate rheology (temperature and frequency sweeps). It is important that they characterized TPU PSAs by tack measurement, creep test and 180º peel test. In this way, this study also describes the moderately improved properties of TPU PSAs. I, therefore, recommend that this manuscript can be published as it is or by addressing some minor issues given below;
TPUs blends showed two Tgs. However, the second Tg is not obvious in the DSC curve. It would be great If authors could add DMA results to show the clear Tg. Authors also mentioned that TPUs blends produce crystalline structure, but DSC curve does not show any crystalline peak and melting peak. Any evidence for the formation of hydrogen bonding would be great.Author Response
See attached file

Reviewer 3 Report
The article is an interesting comparison of methods for obtaining adhesives made from polyurethane mixtures and an analysis of their properties. The article can be published in the journal Polymers after making some corrections by the Authors,
My comments and suggestions:
The authors in the introduction section discussed the relationship between the composition of adhesive mixtures and their properties. Adhesives have become a popular method of gluing elements in industry along with other techniques, such as welding, soldering and riveting. A lot of research, development and engineering work has been carried out to develop the best adhesive technologies. PSA adhesives based on acrylic, silicones, and rarely polyurethanes are popular. Therefore, the choice of research topic regarding adhesives obtained from polyurethane mixtures is very reasonable.
Materials and methods section are detailed. The authors thoroughly characterized all materials. They provided data on the used devices and the conditions under which the tests were carried out. This section is detailed enough to allow you to repeat the entire experiment.
Further, the authors presented the method of conducting research, and then performed a detailed comparative characteristics of adhesive mixtures obtained by selected methods. They did it in the right way. Performing a series of tests to assess the properties of the obtained adhesives. They carried out a correct, in-depth analysis of the obtained test results. However, the obtained results should be compared to the results obtained by other authors who worked on similar solutions and to commercial adhesive products on the market. In this way, the Authors could highlight the advantages of their solution and prove knowledge of not only scientific but also industry literature. This supplement is necessary in this article. I am also asking the authors to indicate in the article the possibilities of using the obtained adhesives.
Round 2
Reviewer 1 Report
The authors responded adequatelly to the majority of the Queries. However I have still three queries for deeper discussion.
1. In order to keep the quality of the Polymers (MDPI) journal and with the respect to the publications of Pistor et. al. published in Journal of Nanomaterials (Q3), it would be better to avoid the presence of the XRD figure as well as discussion as a confirmation of crystalline phase presence. If the crystalline phase is somewhere present the Müller Plains of the diffraction peak have to be assigned correctly as it respect certain crystalline ordering in the materials (similarly as is i.e. for polypropylene). I agree based on the publication of Mattia and Painter (Macromolecules 2007) that investigated samples can have certain ordered structures based on hydrogen bonding may be called "crystalline phase" base on FTIR not XRD, but the statement should be as strong as they are using in the publication. Moreover, they finally showed certain endothermic peak for blend 60/40 after annealing, and eventhough they statements were "would suggest that there is only partial ordering or the size of any crystalline domains is very small". Please also add such DSC study for annealed blend where you expecting this behaviour. I would like to see similar endotherm confirming that crystalline phase is present and upon certain conditions can be highlighted for example to SI.
2. In authors response, authors are claiming that rheometer is working at the temperature range from -10°C to 120°C. Well, using such temperature range and rotational rheometer, it is possible to perform temperature sweep measurement and based on the storage modulus (drops) but more precisely in tan delta (peaks) dependence on temperature such two Tgs have to be clearly visible as double peek or shoulder of the main peak. If peaks are there, just put it to SI. If there are no peaks, the second Tg need to be dismissed from whole article.
3. Finally, the manuscript is missing the information of the state of the art in the filed of pressure sensitive adhesives. Please enlarge the introdution section or add table with corresponding quantities to compare your results with other systems not based on polyurethanes.
